# Revisiting Classical Issues of Fatigue Crack Growth Using a Non-Linear Approach

**DOI:** 10.3390/ma13235544

**Published:** 2020-12-04

**Authors:** Micael F. Borges, Diogo M. Neto, Fernando V. Antunes

**Affiliations:** Department of Mechanical Engineering, Centre for Mechanical Engineering, Materials and Processes (CEMMPRE), University of Coimbra, 3030-788 Coimbra, Portugal; micaelfriasborges@outlook.pt (M.F.B.); diogo.neto@dem.uc.pt (D.M.N.)

**Keywords:** fatigue crack growth, constant amplitude loading, crack closure, overload

## Abstract

Fatigue crack growth (FCG) has been studied for decades; however, several aspects are still objects of controversy. The objective here is to discuss different issues, using a numerical approach based on crack tip plastic strain, assuming that FCG is driven by crack tip deformation. ΔK was found to control cyclic plastic deformation at the crack tip, while *K*_max_ has no effect. Therefore, alternative mechanisms are required to justify models based on Δ*K* and *K*_max_. The analysis of crack tip plastic deformation also showed that there is crack tip damage below crack closure. Therefore, the definition of an effective load range Δ*K*_eff_ = *K*_max_ − *K*_open_ is not correct, because the portion of load range below opening also contributes to FCG. Below crack closure, damage occurs during unloading while during loading the crack tip deformation is elastic. However, if the maximum load is decreased below the elastic limit, which corresponds to the transition between elastic and elasto–plastic regimes, there is no crack tip damage. Additionally, a significant effect of the crack ligament on crack closure was found in tests with different crack lengths and the same Δ*K*. Finally, the analysis of FCG after an overload with and without contact of crack flanks showed that the typical variation of *da/dN* observed is linked to crack closure variations, while the residual stresses ahead of crack tip are not affected by the contact of crack flanks.

## 1. Introduction

Fatigue crack growth (FCG) has been studied for decades; however, several aspects are still objects of controversy. The crack closure concept, proposed by Elber [1], has been widely used to explain the effect of stress ratio, thickness and variable amplitude loading [2,3]. The contact of crack flanks is consensual, since it has been observed using analytical, numerical and experimental approaches, namely digital image correlation [4], X-ray diffraction [4,5], potential drop [6,7] and SEM [7]. However, its effect on FCG is a subject of great discussion. In fact, several researchers have questioned the relevance of crack closure, particularly under plane strain conditions [8,9,10,11,12,13]. These authors claim that FCG trends may be explained without the use of the crack closure concept, and propose alternative approaches, namely the UNIGROW life prediction method [14] and the unified approach [15]. The crack closure defenders suggest that Δ*K*_eff_ is the crack driving force, while the opponents claim that both Δ*K* and *K*_max_ are the driving parameters. González et al. [12] presented the results of *da/dN* under constant *K*_max_ and Δ*K* and obtained constant *da/dN*, as expected; however, crack closure decreased with crack growth. They claimed that the crack opening load is highly dependent on the residual ligament and therefore cannot explain the constant *da/dN*.

The procedures followed to quantify crack closure and fatigue threshold are also controversial. Round robin tests were organized by the ASTM Task Group E24.04.04 in order to compare the conventional closure measurements. Although the material and specimen geometries were the same, a significant influence of laboratory, investigator and technique used was found. It was concluded that “scatter of this magnitude would make it very difficult to develop a clear picture of closure effects and to verify quantitative models of closure effects using data from the literature” [16]. Other issues include the relative importance of residual stresses ahead of the crack tip and the crack closure behind it, and the occurrence of damage below the crack opening load. Those working with crack closure propose an effective value, which is the range between maximum and opening loads. On the other hand, other researchers have questioned the existence of crack closure damage below crack opening [17,18]. There is however a general agreement about the complexity of crack tip phenomena, involving different mechanisms that depend on the material, geometry, and loading. These mechanisms include crack closure, residual stresses, crack-tip blunting, crack branching, phase transformation, and environmental damage. Crack closure may be produced by residual plastic deformation, oxides, or roughness. Cyclic plastic deformation is usually assumed to be the crack driving force; however, environmental damage is supposed to have a significant contribution near-threshold [19].

FCG is usually studied using Δ*K* as a fundamental parameter. Its validity, which is limited to small scale yielding at the crack tip, is not normally tested. In addition, too much importance is given to crack closure and its measurement. In fact, the focus must be placed on the crack tip, where the damage responsible for FCG effectively occurs. Other phenomena, such as crack closure, residual stresses, or material hardening, are secondary but are relevant insofar as they affect the main phenomenon. A numerical approach is followed here to predict the FCG rate, based on the cumulative plastic strain at the crack tip. It is assumed that cyclic plastic deformation is the fundamental mechanism responsible for FCG. This approach includes not only the effects of cyclic plastic deformation, but also crack tip blunting, material hardening and plasticity-induced crack closure. Different classical issues of FCG are revisited, namely the effect of maximum and minimum loads, and the existence of damage below crack opening.

## 2. Material Model

The material studied was the 2024-T351 aluminum alloy. It has good fatigue strength, low environmental impact, and high specific properties, being adequate for transport related industries. In fact, the 2024 alloy with different heat treatments is the most used aluminum alloy in the aircraft industry. The mechanical behavior of this alloy is described by a phenomenological elastic–plastic constitutive model. The isotropic elastic behavior is given by the generalized Hooke’s law, where the adopted Young’s modulus and the Poisson coefficient are listed in Table 1. Regarding the plastic behavior, the shape of the yield surface is defined by the von Mises yield criterion with an associated flow rule. The evolution of the yield surface during plastic deformation is described by the Swift isotropic hardening law combined with the kinematic hardening law proposed by Armstrong–Frederick [20].

The flow stress defined according to the Swift hardening law is given by:(1)$$Y\left({\bar{\varepsilon }}^{p}\right)=K{\left({\left(\frac{Y_{0}}{K}\right)}^{\frac{1}{n}}+{\bar{\varepsilon }}^{p}\right)}^{n}$$
where *Y*_0_, *K*, and *n* are the material parameters of the Swift law and ${\bar{\varepsilon }}^{p}$ is the equivalent plastic strain. The isotropic work hardening law is combined with the kinematic hardening law proposed by Armstrong–Frederick, where the rate of back stress tensor is defined by:(2)$$\dot{X}=C_{X}\left[\frac{X_{\mathrm{sat}}}{\bar{\sigma }}\left(\sigma^{\prime }-X\right)\right]{\dot{\bar{\epsilon }}}_{\mathrm{pl}},\;\mathrm{with}\;\dot{X}\left(0\right)=0$$
where X is the back stress tensor, $X_{Sat}$ and $C_{x}$ are material parameters, σ′ is the deviatoric component of the Cauchy stress tensor, $\bar{\sigma }$ is the equivalent stress, and ${\dot{\bar{\varepsilon }}}^{p}$ is the equivalent plastic strain rate. The isotropic and kinematic hardening parameters were simultaneously calibrated using the stress–strain curves obtained in smooth specimens of the experimental low cycle fatigue tests. Table 1 presents the list of parameters that define the hardening behavior of this aluminum alloy, which were previously obtained in [21].

## 3. Finite Element Model 

The numerical analysis of the fatigue crack growth (FCG) was performed with the in-house finite element code DD3IMP [22]. This software was originally developed to study deep-drawing, and it was adapted to the study of FCG considering the excellent capabilities for the modeling of plastic deformation in metals. Compact tension (CT) specimens are adopted in this study, whose geometry and main dimensions are shown in Figure 1a. The initial crack size is *a*_0_ = 19 mm in all loading cases analyzed in this study. Due to the geometrical and loading symmetry in relation to the plane of the crack, only the half specimen was modelled considering adequate boundary conditions. Plane stress conditions are considered in the numerical model. Thus, the thickness of the specimen used in the numerical simulations was reduced to 0.1 mm. Figure 1b shows the plane stress boundary conditions.

A constant amplitude cyclic load was applied remotely in the periphery of the specimen hole. The loading is pure mode I since the load is normal to the crack growth direction. The maximum and minimum values of the loads adopted in the numerical simulation are listed in Table 2 for seven different loading cases, which are divided into two groups. In the first group composed of four different loading cases, the minimum force is kept constant (*F*_min_ = 0 N) while the maximum force is variable. On the other hand, the second group is composed of the loading cases where the maximum force is kept constant (*F*_max_ = 50 N) while the minimum force is variable, leading to different values of the stress ratio.

The finite element mesh of the CT specimen is composed of 7287 isoparametric hexahedral finite elements and 14,918 nodes. In order to reduce the computational cost, only the region around the crack tip is defined by a refined mesh with an edge element size of 8 μm. Hence, the severe stress gradients arising in this zone can be accurately predicted. The plasticity-induced crack closure mechanism is considered by modelling the contact between the crack flanks. Taking into account the symmetry conditions, the frictionless contact is established between the crack flank and a stationary rigid surface. On the other hand, the crack closure can be disabled by removing the rigid surface and consequently allowing the crack flanks to overlap.

In this study, the fatigue crack growth is modelled by nodal release, using the approach proposed in [23]. The crack propagation is uniform along the thickness, simultaneously releasing both crack front nodes. The nodal release occurs when the plastic strain at the crack tip achieves a critical value. Nevertheless, it is only performed when the load is at minimum to avoid eventual convergence problems related to the high tensile stresses occurring at maximum load. Assuming that the damage accumulation is responsible for FCG, the total plastic strain accumulated during the entire cyclic loading is considered. Only a single material parameter is required for this fatigue crack growth criterion, which simplifies its usage. Accordingly, the critical value of plastic strain involved in this FCG criterion was calibrated for this aluminum alloy in a previous work, comparing experimental *da/dN* values with numerical predictions under plane stress conditions [21]. In this study, the FCG rate (*da/dN*) is assessed through the ratio between the element size (8 μm) and the number of load cycles required to achieve the critical value of plastic strain at the crack tip.

## 4. Results

### 4.1. Effect of Maximum and Minimum Loads on FCG Rate

Figure 2a shows the variation of *da/dN* with Δ*K* for the two load sets with fixed values of maximum and minimum loads. In both cases, there is an increase in *da/dN* with Δ*K*, as could be expected. However, there is an effect of *K*_max_, since different crack growth rates are obtained for the same Δ*K*. The difference between both sets is higher for lower values of Δ*K*, because the variation of *K*_max_ increases with the decrease in load range.

Figure 2b plots *da/dN* versus effective Δ*K*, being Δ*K*_eff_ = *K*_max_ − *K*_open_. The crack opening level was measured at the first node behind the crack tip. The results from both sets are now nearly coincident, indicating that the crack closure is able to accommodate the effects of maximum and minimum loads on FCG rate. Considering both sets, a Paris law exponent of 3.3 was obtained, with a correlation factor of 0.997. The results of *da/dN* versus Δ*K* are also presented in Figure 2b, showing that the use of Δ*K*_eff_ translates the curves to the left side. This is logical since Δ*K*_eff_ is lower than Δ*K*.

Figure 3 plots *da/dN* − Δ*K* curves obtained without the contact of the crack flanks. The two sets are overlapped, which indicates that *K*_max_ has no effect on cyclic plastic deformation at the crack tip. A similar trend was obtained in Ti6-Al4-V, which did not show the infuence of the stress ratio on *da/dN* [23]. The effect of *K*_max_ observed in Figure 2a is therefore linked with variations of the crack closure phenomenon.

However, there are alternative approaches assuming a two-parameter driving force based on *K*_max_ and Δ*K*. Kujawski [24] proposed for the crack driving force the parameter Δ*K*_eff_
*= (K*_max_*)*^α^ × *(**Δ**K^+^)*^1−^^α^, which is a function of *K*_max_ and Δ*K^+^*, the positive range of Δ*K.* The sensitivity to the *K*_max_ value is quantified by the α parameter. Values of 0.6, 0.5, 0.5, 0.33 and 0.3 were calculated for austempered ductile iron, AA2024-T351, AA7075-T6, Udimet 720 nickel base superalloy and medium carbon steel, respectively. Therefore, according this approach, materials that are usually assumed to have a ductile behaviour are partially brittle. Llanes et al. [25] considered that *K*_max_ is the main driving force for FCG in WC-Co cemented carbides, which was attributed to the predominance of static failure modes. They proposed that *da/dN* = *C* × (*K*_max)_^n^ × (Δ*K*)^m^, where the values *n* of and *m* quantify the relative dominance of each parameter. Sadananda et al. [10,15] and Glinka et al. [11,14] also claimed that *K*_max_ and Δ*K* are sufficient to account for the material response. Sunder [19] proposed the relevance of crack tip stress, σ_tip_, near-threshold. However, at higher load levels, the same concept may be applied, i.e., the damage at the crack tip is certainly controlled by the maximum stress there. The use of *K*_max_ as crack driving force assumes that there is a perfect correlation of this parameter with maximum stress at the crack tip. However, this is not straightforward since there is a dependence on elastic–plastic material properties. Benz [26] claimed the influence of σ_tip_ in his numerical studies. The use of σ_tip_ to include the effects of alternative mechanisms on the FCG rate may be a solution to unify the approaches proposed by Sunder, Glinka and Sadananda.

Since the cyclic plastic deformation is independent of *K*_max_, alternative damage mechanisms are required to explain the effect of *K*_max_. Possible mechanisms, driven by *K*_max_, are the growth and coalescence of microvoids, diffusion-based mechanisms and brittle failure. The diffusion mechanisms, which include environmental damage and creep, greatly depend on material and temperature. In fact, oxidation is known to be a main mechanism in high temperature fatigue of nickel-base superalloys [27,28]. The coalescence of microvoids is a ductile mechanism, while clevage is the brittle decohesion at crystallographic planes. The identification and quantification of the brittle or ductile mechanisms activated by *K*_max_ are of major importance to understand FCG.

### 4.2. Effect of Minimum Load on Crack Opening Level

Tomas Vojtek [29] questioned the effect of the increase in *K*_min_ on crack opening: “If there is no material damage and no significant plastic deformation of the crack tip occurs below the Kop level, how can *K*_min_ influence any of the processes leading to a change in the plasticity-induced crack closure?”. Figure 4 plots the load patterns and the crack opening levels predicted numerically using the contact status of the first node behind the crack tip. In Figure 4a, the increase in minimum load produces an increase in the crack opening level. A similar trend is observed for the effect of maximum load, i.e., an increase in the opening load with *F*_max_.

Figure 5 presents the crack closure level, quantified by:(3)$$U^{*}=\frac{F_{open}-{\;F}_{min}}{F_{max}-F_{min}}\times 100$$
where *F*_open_ is the crack opening load. This parameter represents the percentage of load cycle where the crack is closed. The values of *U** are relatively small, compared with those typically observed under plane stress conditions (≈40–50%). However, this is not strange since the values of *U** depend on load parameters and material properties. As can be seen, the increase in *K*_min_ (fixing *K*_max_) significantly reduces the crack opening level, which is a classical result. In fact, the increase in stress atio is known to reduce the crack closure phenomenon. For *R* > 0.6, there is no crack closure. On the other hand, the increase in *K*_max_ does not produce a well-defined trend. Note that the increase in *K*_max_ is accomplished by an increase in Δ*K*, which has opposite effects on the crack closure level.

The crack closure level depends on three mechanisms: monotonic plastic deformation, reversed plastic deformation and crack tip blunting. The residual plastic wake, i.e., the set of residual plastic wedges behind the crack tip, greatly depends on the maximum load. The increase in maximum load increases the crack tip plastic deformation and therefore the crack closure level. On the other hand, the reversed plastic deformation reduces the elongation of residual plastic wedges, reducing crack closure. The material behaviour, namely the isotropic and kinematic components of hardening, play a major role in these phenomena. In Figure 6a, the difference between the positions of crack flank for an elastic behaviour and for an elasto–plastic behaviour is the elongation of residual plastic wedge (Δ*y*_p_). Crack tip blunting is less evident and less known since it has a subtle effect on the opening level. The increase in load produces monotonic plastic deformation but also crack tip blunting, and this reduces the impact of residual plastic wedges on crack tip stress and strain fields. Blunting has an immediate effect on plasticity-induced crack closure, contrarily to the size of the plastic wedge, which needs propagation. This mechanism is particularly evident in the case of overloads, producing a dramatic effect of crack closure level, usually eliminating it completely. This crack tip blunting mechanism is according to Sadananda’s proposition that plasticity opens the crack rather than closes it [30]. The increase in minimum load, fixing the maximum load, changes the complex process of crack tip plastic deformation. The assumption that the increase in minimum load could occur without the change of opening load, for example, is an oversimplified analysis. Figure 6b plots the crack opening level versus the size of residual plastic wake, Δ*y*_p_. There is an increase in *F*_open_ with Δ*y*_p_, as could be expected. However, there is a deviation from linear proportionality, which can be attributed to crack tip blunting, which increases with load level.

However, the behaviour changes substantially with compressive loads. Two additional load cases were considered with minimum loads of −20 and −40 N, keeping the maximum load at 50 N. Figure 7 plots the crack opening load versus the stress ratio (=*F*_min_/*F*_max_). As can be seen, when the minimum load is compressive, the variation of crack opening level is relatively small. This can be attributed to a saturation of reversed plastic deformation. A detailed analysis of plasticity-induced crack closure under compressive loads can be found in Antunes et al. [31].

### 4.3. Are There Things Happening While the Crack Is Closed?

There is a discussion about the occurrence of damage while the crack is closed. According to Elber’s concept of crack closure, the contact of crack flanks eliminates the damage at the crack tip, which justifies the use of Δ*K*_eff_. Figure 8a plots crack tip opening displacement (CTOD), measured at a distance of 8 μm behind the crack tip, versus applied force. There is a small difference between the crack closure and crack opening loads, points F and B, respectively, which is a clear indication that something happens (irreversible) while the crack is closed. If nothing happens, there is no reason for the difference between the crack closure and crack opening.

Figure 8b shows the cumulative plastic strain measured at the crack tip. There is plastic deformation increase in the segment FG, during which the crack is closed. Note that the deformation in segment FG is only 5.4% of the total plastic deformation. This deformation occurs progressively up to the minimum load. On the other hand, during loading (segment AB), there is no plastic deformation and only elastic deformation. This is very important because it clearly indicates that the definition of an effective load range as Δ*K*_eff_ = *K*_max_ − *K*_open_ is not correct, because the portion of load range below the crack opening also contributes to FCG. Elber’s hypothesis that the FCG driving force is Δ*K*_eff_ implies that there is no activity ahead of the crack tip for *K* < *K*_op_ [12]. Professor Ravi Chandran says [29]: “I should also point out that any notion of crack tip deformation at *K* < *K*_open_ fundamentally violates the definition of ∆*K*_eff_”. Vojtek et al. [8] also pointed out that Δ*K*_eff_ may not be a good parameter for the quantification of the crack driving force, since the relationship between *K*_max_ − *K*_cl_ and the cyclic plastic deformation at the crack tip might not be linear.

These results totally agree with Professor Daniel Lingenfelser [29]: “However, I disagree with your assumption that there is ‘no material damage and no significant plastic deformation of the crack tip occurs below the Kop level’. I agree that most of the ‘damage’ occurs above *K*_op_ but plastic yielding occurs in compression around the crack/notch tip during the unloading part of the cycle. This plastic deformation when the crack is closing determines the *K*_op_ for the next cycle. Therefore, changing *K*_min_ will cause a change to *K*_op_”.

Figure 9a plots the variation of crack tip stress during one load cycle. As can be seen, σ_yy_ stress is always changing, particularly while the crack is open. Below the crack closure and crack opening levels, there is some variation of stress and therefore of strain, but that is less relevant. During loading from minimum load, the deformation is purely elastic, as is observed in Figure 8b. Figure 9b plots the percentage of plastic deformation observed below the closure relatively to the plastic deformation accumulated during the complete load cycle for the load cases presented in Table 2. The values obtained range from 2.4% to 13.3%.

Another test was undertaken, as illustrated in Figure 10a. Crack was propagated in order to generate residual plastic wake and crack closure. After that, the maximum load was reduced below the opening load, as illustrated. The objective is to check if this load cycling below the crack opening produces damage at the crack tip and therefore FCG. Figure 10b shows the evolution of cumulative plastic strain. As can be seen, after the load reduction, identified by the vertical line, there is no accumulation of damage at the crack tip; therefore, the crack does not propagate. A similar result was obtained when the maximum load was above the opening load, but below the elastic limit, defined by point C in Figure 8a.

### 4.4. Effect of Crack Ligament

Another issue is the effect of crack ligament on the crack closure level and FCG. New constant amplitude load cases were defined, as indicated in Table 3, with different crack lengths. The objective was to have the same Δ*K* for different crack lengths. Two load sets were defined, having constant *F*_min_ or constant *F*_max_. There are small variations of Δ*K* because the numerical tests had a fixed number of applied load cycles, which corresponded to different final crack lengths. As can be seen in Figure 11a, although Δ*K* is constant, there is a significant variation of *da/dN*, particularly for the set with constant *F*_max_. The increase in the FCG rate is associated with the decrease in crack closure level. Figure 11b plots *da/dN* versus Δ*K*_eff_. The results of Figure 2b are also presented, and a perfect alignment can be observed. This indicates that the variation of *da/dN* observed in Figure 11a is due to crack closure variations. This variation of crack closure, which was observed by other authors [12], is important for the understanding of the FCG phenomenon. It is probably linked to variations of crack tip blunting. 

### 4.5. Residual Stresses versus Crack Closure

Another controversy is the relative importance of residual stresses and crack closure on the FCG rate. The researchers that subscribe to crack closure as the main mechanism claim that the effect of an overload is linked with crack closure variations. On the other hand, there is a significant group of researchers denying the relevance of crack closure and proposing alternative concepts to explain different trends of the FCG rate. Figure 12 plots the effect of an overload of *OLR* = 1.5 on *da/dN*, with the overload ratio defined as:(4)$$OLR=\frac{F_{OL}-F_{min}}{F_{max}-F_{min}}\times 100$$

Two models were defined, with and without contact of crack flanks, and the results are presented in Figure 12a,b, respectively. The results in Figure 12a show the typical behaviour of an overload. There is a peak of *da/dN,* which is associated with the elimination of crack closure due to crack tip blunting. In fact, this blunting separates crack flanks, eliminating the effect of residual plastic wake illustrated in Figure 6a. The growth ahead of the overload application point rapidly produces crack closure, which is stronger than under constant amplitude loading. This explains the decrease in *da/dN* to a minimum value. The minimum value occurs after some crack growth, and this phenomenon is called delayed retardation. After that, there is a progressive increase to the value of *da/dN* corresponding to constant amplitude loading. In fact, as the crack propagates, the plastic deformation associated with the overload progressively moves backward relatively to the crack tip. In other words, the effect of the overload progressively disappears as the crack propagates. The constant amplitude loading presents a decrease in *da/dN* at the beginning of crack propagation, which is associated with the formation of residual plastic wake.

On the other hand, the results of Figure 12b are radically different. The overload produces a local peak and nothing else. There is no delay of the crack growth. Therefore, the effect of an overload must be associated with the crack closure phenomenon. The constant amplitude loading presents a minor oscillation at the beginning of crack propagation, which is due to the stabilization of cyclic deformation. After that, there is a slow but progressive increase in *da/dN* with crack growth, which is due to the increase in crack tip stress associated with crack length. Figure 13 plots the residual stresses immediately after the overload, once again with and without the contact of crack flanks. As can be seen, the residual stresses ahead of the crack tip were not affected by the elimination of contact and therefore cannot explain the dramatic modification of *da/dN* − Δ*a* plots.

## 5. Conclusions

Fatigue crack growth has been studied for decades; however, several issues are still controversial. In this paper, some of these issues are revised using a numerical prediction approach based on cumulative plastic strain at the crack tip. 

The main conclusions are:*K*_max_ has no effect on cyclic plastic deformation at the crack tip. Therefore, the approaches assuming a two-parameter driving force based on *K*_max_ and Δ*K* are implicitly proposing other crack tip damage mechanisms;While the crack is closed, there is an increase in plastic deformation, but only during unloading. During loading from the minimum load to the crack opening load, there is no plastic deformation. The deformation during unloading was found to be in the range between 2.4% to 13.3% of the total plastic deformation. This is relevant because it invalidates the classical definition of an effective load range as Δ*K*_eff_ = *K*_max_ − *K*_open_;The variation of *da/dN* produced by an overload was clearly associated with crack closure. In fact, the delay of crack growth vanishes when the contact of crack flanks is eliminated. On the other hand, the residual stresses ahead of the crack tip were not affected by the elimination of contact and therefore cannot explain the dramatic modification of *da/dN* − Δ*a* plots.

Other issues, such as the effect of the environment [32], mixed mode loading [33] or the uncertainty of FCG [34], are addressed for future work.

## Figures and Tables

**Figure 1 materials-13-05544-f001:**
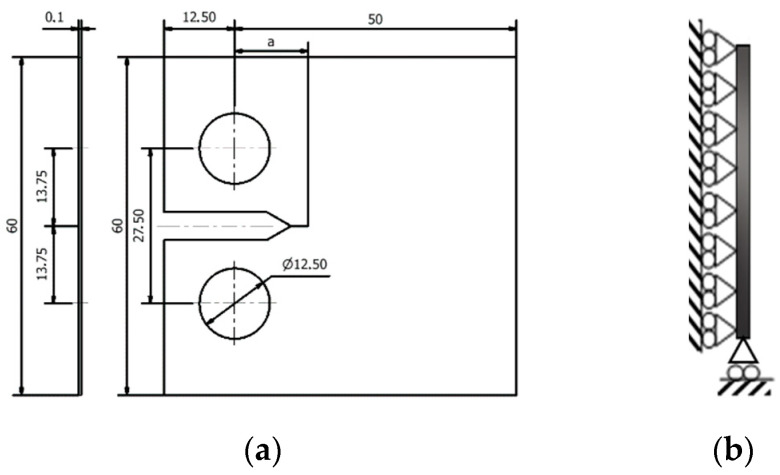
(**a**) Compact tension (CT) specimen modelled for AA2024-T351, with dimensions in mm. (**b**) Plane stress boundary conditions.

**Figure 2 materials-13-05544-f002:**
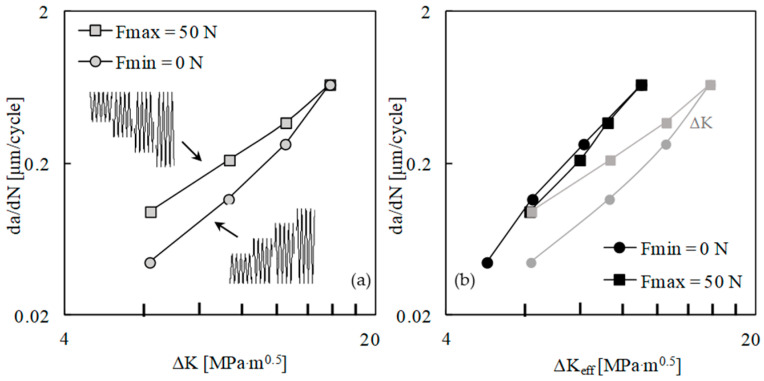
Effect of the constant amplitude loading limits on the fatigue crack growth (FCG) rate: (**a**) *da/dN* versus Δ*K*. (**b**) *da/dN* versus Δ*K*_eff_.

**Figure 3 materials-13-05544-f003:**
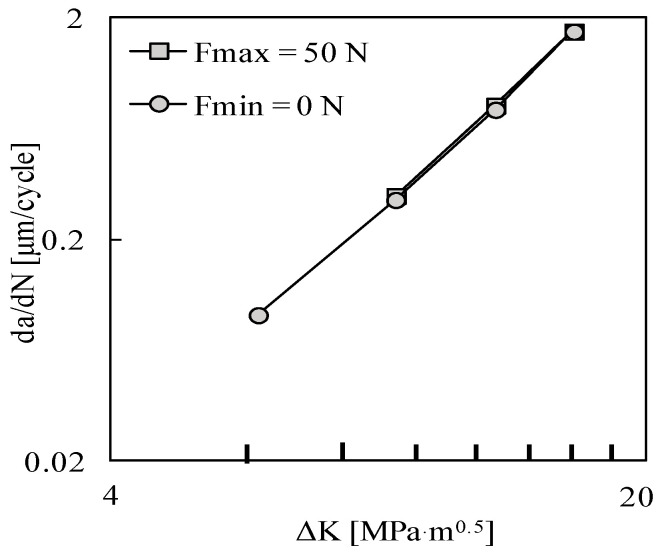
Effect of the constant amplitude loading limits on the FCG rate when the contact of crack flanks is neglected: *da/dN* versus Δ*K*.

**Figure 4 materials-13-05544-f004:**
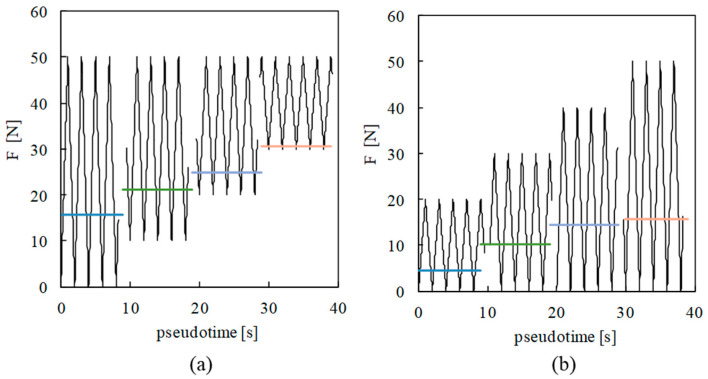
Predicted crack opening load: (**a**) fixing the maximum load, *F*_max_. (**b**) Fixing the minimum load, *F*_min_.

**Figure 5 materials-13-05544-f005:**
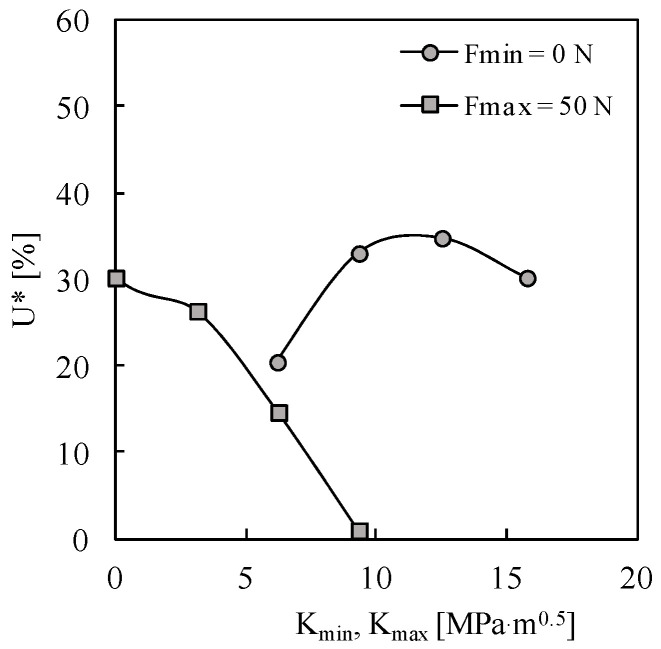
Variation of crack opening parameter with maximum and minimum loads.

**Figure 6 materials-13-05544-f006:**
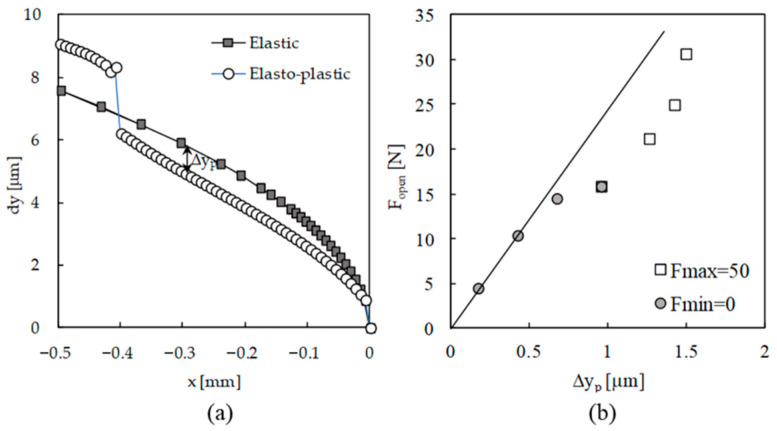
(**a**) Elongation of residual plastic wedge, Δ*y*_p_, measured at maximum load (*F*_max_ = 50 N; *F*_min_ = 0). (**b**) Crack opening level versus Δ*y*_p_.

**Figure 7 materials-13-05544-f007:**
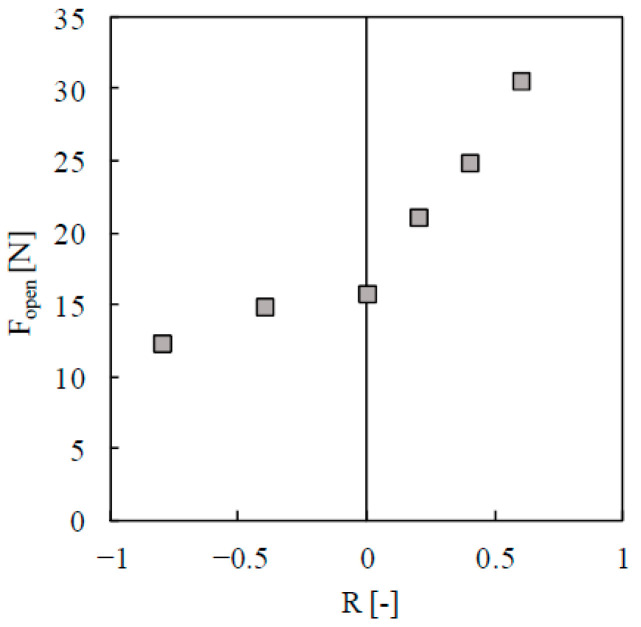
Crack opening load versus stress ratio, *R* for the loading cases with fixed maximum load (*F*_max_ = 50 N).

**Figure 8 materials-13-05544-f008:**
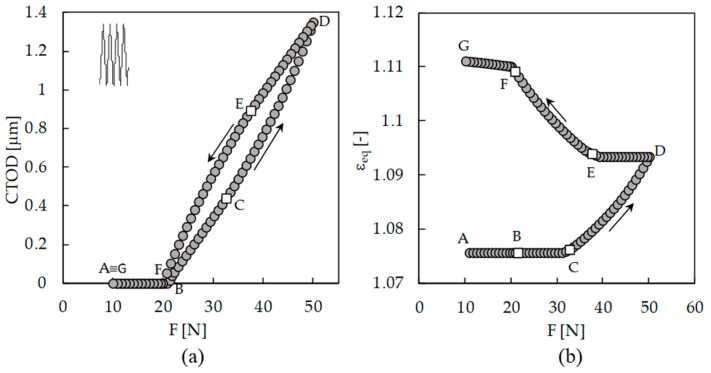
(**a**) CTOD versus load. (**b**) Cumulative plastic strain at crack tip versus applied load (*F*_max_ = 50 N; *F*_min_ = 10 N).

**Figure 9 materials-13-05544-f009:**
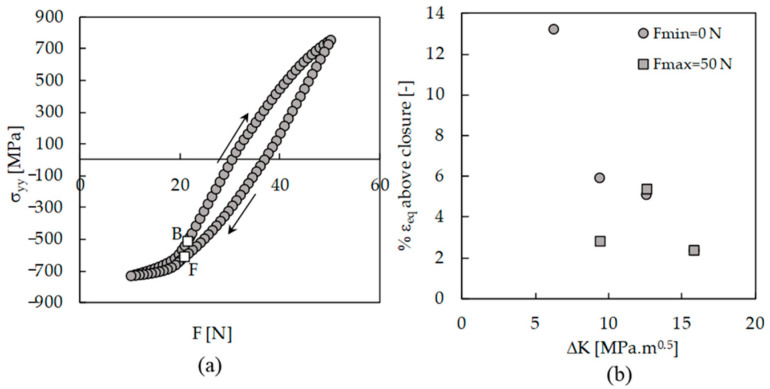
(**a**) Crack tip stress versus remote load (*F*_max_ = 50 N; *F*_min_ = 10 N). (**b**) Percentage of deformation below closure.

**Figure 10 materials-13-05544-f010:**
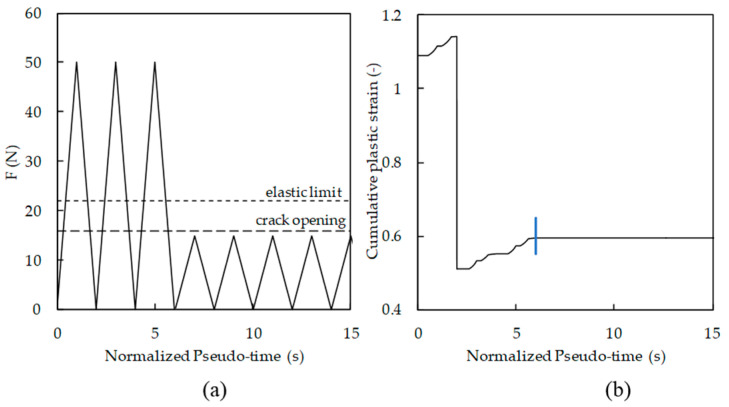
Effect of high–low load sequence (**a**) Spectrum of the applied load. (**b**) Evolution of the cumulative plastic strain at the crack tip.

**Figure 11 materials-13-05544-f011:**
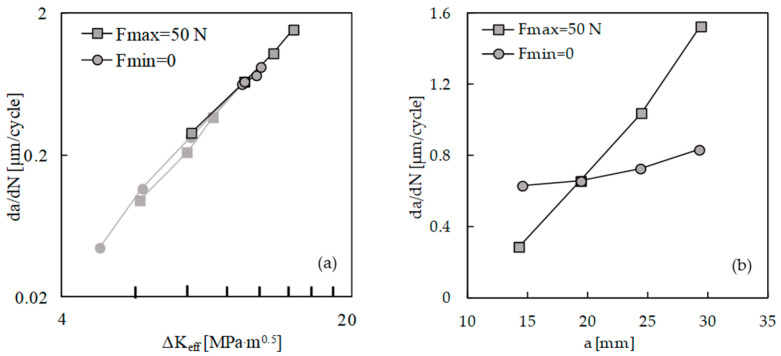
Loading cases with constant Δ*K*: (**a**) *da/dN* versus a. (**b**) *da/dN* versus Δ*K*_eff_.

**Figure 12 materials-13-05544-f012:**
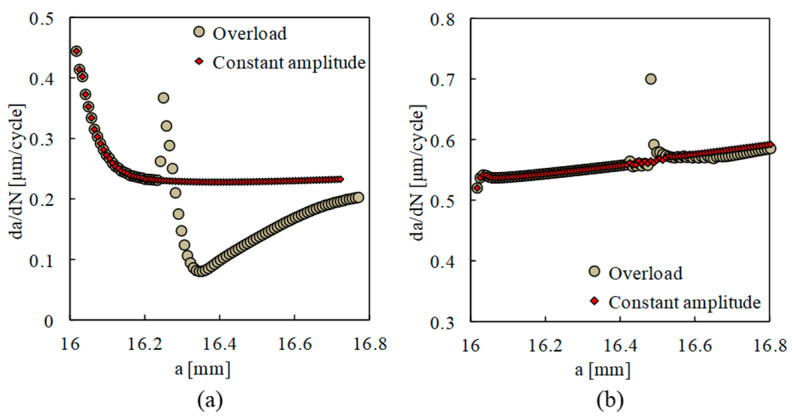
Effect of an overload on *da/dN*. (**a**) With contact of crack flanks. (**b**) Without contact of crack flanks (*F*_max_ = 41.67 N; *F*_min_ = 4.17 N; *F*_OL_ = 60.42 N; Δ*K*_BL_ = 10.3 MPa·m^0.5^; *OLR* = 1.5).

**Figure 13 materials-13-05544-f013:**
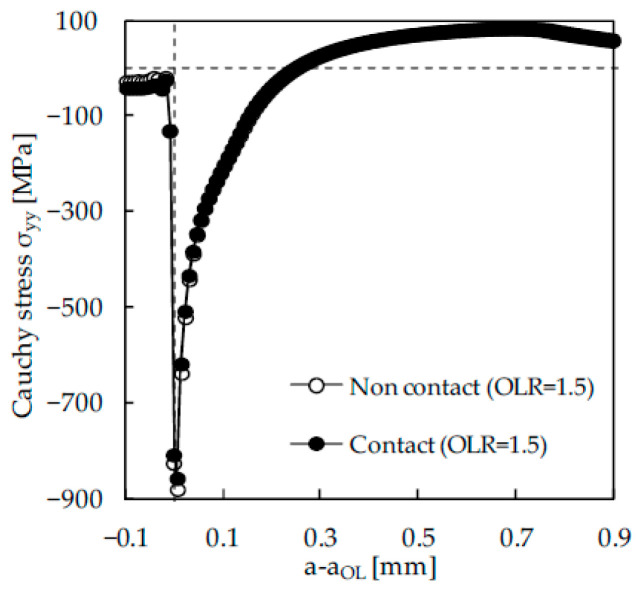
Contact stresses at minimum load immediately after the application of an overload (*F*_max_ = 41.67 N; *F*_min_ = 4.17 N; *F*_OL_ = 60.42 N; Δ*K*_BL_ = 10.3 MPa·m^0.5^; *OLR* = 1.5).

**Table 1 materials-13-05544-t001:** Elastic properties of 2024-T351 aluminum alloy and parameters obtained for the Swift isotropic hardening law combined with the Armstrong–Frederick kinematic hardening law.

Material	*E* (GPa)	*ν* (-)	*Y*_0_ (MPa)	*K* (MPa)	*n* (-)	*X*_Sat_ (MPa)	*C*_X_ (-)
AA2024-T351	72.26	0.29	288.96	389.00	0.056	111.84	138.80

**Table 2 materials-13-05544-t002:** Different loading cases under constant amplitude for the CT specimen with initial crack size *a*_0_ = 19 mm.

Load Case	*F*_min_ (N)	*F*_max_ (N)	*K*_min_ (MPa·m^0.5^)	*K*_max_ (MPa·m^0.5^)	*R*
1	0	20	0	6.2	0
2	0	30	0	9.3	0
3	0	40	0	12.5	0
4	0	50	0	15.8	0
5	10	50	3.1	15.7	0.2
6	20	50	6.2	15.6	0.4
7	30	50	9.3	15.6	0.6

**Table 3 materials-13-05544-t003:** Loading cases with different crack size and nearly constant Δ*K*.

Crack Length	*F*_min_ (N)	*F*_max_ (N)	Δ*K* (MPa·m^0.5^)	*F*_open_ (N)	*U** (%)	*da/dN* (μm/cycle)
14.544	0	65.4	16.06	22.1	32.3	0.63
19.408	0	50	15.78	15.8	30.2	0.66
24.352	0	38.3	15.92	10.6	26.1	0.73
29.24	0	27.5	15.82	6.9	23.8	0.83
14.264	−15.44	50	15.83	16.8	48.2	0.29
19.408	0	50	15.78	15.8	30.2	0.66
24.432	11.7	50	16.00	19.6	19.1	1.04
29.376	22.5	50	15.99	25.6	9.5	1.53

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
