# Peer review of "Revisiting Classical Issues of Fatigue Crack Growth Using a Non-Linear Approach"

_materials, 2020, doi:10.3390/ma13235544_

Round 1

Reviewer 1 Report

Thank You for this interesting article. The paper discussess many topics related to the crack growth and fatigue life prediction. It describes the mechanisms behind cyclic and overload loading programs and the effect of varius mean stress values described with appropriate R- ratios. Nevertheless it would be interesting to discuss variable amplitude loading or at least mention some of the mechanisms that are causing the initiation or uncertaintity as presented inter alia in the papers:

10.3390/ma13020423

10.3390/ma13010160

10.3390/met10050646

Author Response

Reviewer 1

Thank You for this interesting article. The paper discussess many topics related to the crack growth and fatigue life prediction. It describes the mechanisms behind cyclic and overload loading programs and the effect of varius mean stress values described with appropriate R- ratios. Nevertheless it would be interesting to discuss variable amplitude loading or at least mention some of the mechanisms that are causing the initiation or uncertaintity as presented inter alia in the papers:

Authors: The study of variable amplitude loading is very interesting for a better understanding of FCG. In this paper we only study the effect of overloads, but we have new numerical tests running to study the effect of load blocks and a more complex load pattern called super block. In the future we intend to study more realistic load patterns.

It is our goal to continue this line of research once we believe in its potential. For future work we are planning the inclusion of uncertainty in the approach. The numerical approach proposed is deterministic, however we are aware that there is always some uncertainty associated with small perturbations of material properties, loading and geometry.

Three new references were added, as recommended:

Toribio, J.; Matos, J.-C.; González, B. Corrosion-Fatigue Crack Growth in Plates: A Model Based on the Paris Law, Materials 2017, 10, 439.

Akama, M.; Kiuchi, A. Fatigue Crack Growth under Non-Proportional Mixed Mode Loading in Rail and Wheel Steel Part 2: Sequential Mode I and Mode III Loading, Appl. Sci. 2019, 9(14), 2866.

Chang,  H.; Shen, M.; Yang, X.; Hou, J. Uncertainty Modeling of Fatigue Crack Growth and Probabilistic Life Prediction for Welded Joints of Nuclear Stainless Steel, Materials 2020, 13, 3192.

Reviewer 2 Report

Analysis of fatigue crack growth in the engineering industry plays a very important role. Many factors affect the formation and propagation of cracks. The importance of predicting the occurrence of fatigue cracks in the operation of equipment is often greater than the analysis of defects after accidents. Every new approach, procedure or method that is developed in this direction is interesting and brings new knowledge to the scientific field.

The article is aimed to discuss different issues, using a numerical approach based on crack tip plastic strain, assuming that FCG is driven by crack tip deformation.

The authors compare the results achieved by different authors to find or confirm the correctness of their proposed model.

The studied issue is broad and in my opinion it is not easy to unambiguously determine the degree of correctness of the solutions of other authors only according to the published results.

A more significant contribution to the article are the authors' own analyzes, which they performed while studying the issue.

Analyzes were performed on a specific material, a specific shape of the test specimen at a specific load.

Please let the authors comment on some issues if possible.

Why did the authors choose 2024-T351 aluminum alloy as reference material?

What area of technical practice is this material characteristic of?

Is the software used the authors' own product or is it a commercial software?

Are all the results obtained only from finite element calculations?

Did the authors analyze the behavior of the material at different temperatures?

What are the practical benefits of the findings?

Where will it be possible to apply the research results?

Reviewer 3 Report

The submitted manuscript entitled ‘Revisiting classical issues of fatigue crack growth using a non-linear approach’ is dealing with the analysis of the fatigue crack growth phenomenon using a non-linear approach and a new point of view. The manuscript is interesting and would be worth to publish, the only problem is in the lack of physical measurements. The virtual environment and quite unusual sample geometry (extremely thin) in the case of new consequences and judgements according to the well-known problem of fatigue crack growth are simply not enough convincing.

Round 2

Reviewer 1 Report

The corrected version is suitable for publication.

Reviewer 3 Report

Dear Authors,

Thank you for your answer, eagerly waiting for the experiments, I hope that the results will confirm your simulations.